# AGENT SKILL ACQUISITION FOR LARGE LANGUAGE MODELS VIA CYCLEQD

**So Kuroki, Taishi Nakamura, Takuya Akiba, Yujin Tang**
Sakana AI, Japan
{sokuroki,taishi,takiba,yujintang}@sakana.ai

## ABSTRACT

Training large language models to acquire specific skills remains a challenging endeavor. Conventional training approaches often struggle with data distribution imbalances and inadequacies in objective functions that do not align well with task-specific performance. To address these challenges, we introduce CycleQD, a novel approach that leverages the Quality Diversity framework through a cyclic adaptation of the algorithm, along with a model merging based crossover and an SVD-based mutation. In CycleQD, each task's performance metric is alternated as the quality measure while the others serve as the behavioral characteristics. This cyclic focus on individual tasks allows for concentrated effort on one task at a time, eliminating the need for data ratio tuning and simplifying the design of the objective function. Empirical results from AgentBench indicate that applying CycleQD to LLAMA3-8B-INSTRUCT based models not only enables them to surpass traditional fine-tuning methods in coding, operating systems, and database tasks, but also achieves performance on par with GPT-3.5-TURBO, which potentially contains much more parameters, across these domains. Crucially, this enhanced performance is achieved while retaining robust language capabilities, as evidenced by its performance on widely adopted language benchmark tasks. We highlight the key design choices in CycleQD, detailing how these contribute to its effectiveness. Furthermore, our method is general and can be applied to image segmentation models, highlighting its applicability across different domains[1].

## 1 INTRODUCTION

Large Language Models (LLMs) have established themselves as powerful tools in the machine learning and artificial intelligence landscape, initially gaining prominence through their effectiveness in conversational tasks (Achiam et al., 2023). As the demand grows for LLMs to perform a broader range of cognitive tasks, the ability to integrate linguistic understanding with actionable outputs becomes essential. This integration facilitates the creation of intelligent LLM-based agents, making continual agentic fine-tuning a critical development in the field (Gur et al., 2023; Liu et al., 2023b; Xi et al., 2023; 2024; Wang et al., 2024b). However, training LLMs to acquire various agent skills presents two major challenges that complicate their development.

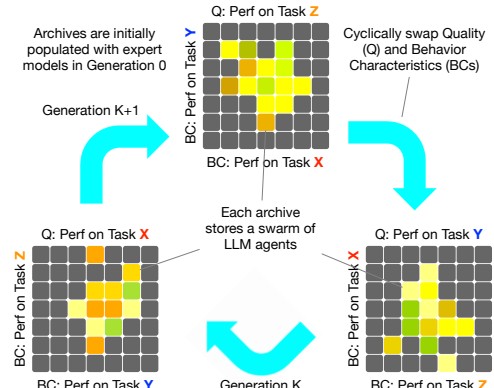

Figure 1: **Method Overview.** CycleQD uses QD in a cyclic manner to merge LLMs. Archives are initialized with expert LLMs, each of which has been fine-tuned to excel in a specific task. Model merging is conducted using QD, treating task performance as quality (Q) and behavior characteristics (BC), which are cyclically swapped in each generation.

---

[1]https://github.com/SakanaAI/CycleQD

A significant difficulty arises from the need to balance data ratios during training, to ensure that the model learns effectively from datasets representing different skills without favoring any particular one and without neglecting others. For example, researchers have identified the significant influence of code training to a model's inferential generation and thinking capabilities (Liu et al., 2023b). Comparing CODELLAMA and LLAMA-2, the former shows a significant edge in tasks that follow a relatively static procedure after being tuned on code data, yet at the same time reveals decreased performance in tasks that require general thinking abilities.

In addition, conventional objective functions, such as next token prediction, often fail to capture the nuances required for performance across diverse tasks, typically leading to sub-optimal training outcomes. This function, although effective for general language understanding, does not foster the development of specific agent skills. An alternative, such as reinforcement learning (RL), has been shown to align LLMs with user intents more effectively: Ouyang et al. (2022) demonstrated aligning LLMs with user intents by RL from human feedback, where a reward model is trained from a dataset of human-ranked model outputs. This method has later on been adopted by many others and become an industrial standard. Fortunately, unlike general language tasks, the "rewards" are clearly defined in most agent tasks. On the other hand, managing the balance of rewards in learning tasks and the associated costs of fine-tuning pose additional problems.

In response to these challenges, we propose CycleQD, a novel framework that leverages the Quality Diversity (QD) method (Pugh et al., 2016), specifically through a cyclic adaptation of the MAP-Elites algorithm (Mouret & Clune, 2015). CycleQD features a dynamic cycle where each agentic skill's performance metric (e.g., pass@1 for coding tasks, success rate for Operation System (OS) and Database (DB) tasks) is optimized in isolation, with other skills represented as behavioral characteristics (BCs). Unlike conventional crossover operations that randomly swap the parameters between two genes, CycleQD employs a model merging based crossover operation (Akiba et al., 2024), which is essential for transferring skills from specialized experts to a new, cohesive model. Furthermore, we also introduce a singular value decomposition (SVD)-based mutation method to extrapolate model capabilities while preventing overfitting. Unlike gradient-based fine-tuning, CycleQD reduces the need for intricate design decisions and hyperparameter tuning, while inherently capturing diverse skills and behaviors. See Figure 1 for an overview of CycleQD.

We adopt design choices specifically targeting the two major challenges in continual agentic fine-tuning. Our proposed method not only simplifies the complex data ratio management by focusing on one task at a time but also addresses the inadequacy of traditional loss functions by directly optimizing task-specific performance metrics. Moreover, the QD framework proves ideal for model merging, particularly in LLMs where the computational pipelines are long. By allowing locally sub-optimal solutions to persist, such as a temporarily under-performing layer, QD saves these configuration in its archive and provides them a chance to improve in the future.

The empirical results from our experiments illustrate that CycleQD significantly enhances the ability of a LLAMA3-8B-INSTRUCT based LLM agent to develop and refine multiple computer science skills. Notably, the averaged performance of the final 8-billion parameter open-weight model is on par with that of GPT-3.5-TURBO on coding, OS and DB tasks, which potentially contains much more parameters. This superior performance is achieved without the typical degradation associated with shifting focus among tasks or compromising general language capabilities.

Based on these results, it is important to note that while post-training of LLMs has been dominated by gradient-based optimizations, our method demonstrates that incorporating evolutionary algorithms can serve as a compelling modification to the conventional fine-tuning pipeline, effectively enhancing the training of LLM agents. Our technical contributions are summarized as the following:

- We introduce CycleQD, a novel approach for merging LLM agents, each specializing in a different task, to create a composite LLM that outperforms baseline methods on computer science tasks from AgentBench (Liu et al., 2023b) and popular coding benchmarks while maintaining language capabilities.

- We provide ablation studies and detailed explanations of the key design choices in CycleQD, illustrating their pivotal roles in enhancing the method's effectiveness.

- We extend the application of CycleQD to the Segment Anything Model (SAM) (Kirillov et al., 2023), demonstrating our method's broad applicability across domains.

## 2   PRELIMINARIES

**Evolutionary Strategies (ES)** are optimization algorithms inspired by natural selection. They improve a population of candidate solutions through mutation, crossover (i.e., recombination), and selection, guided by a fitness function. By maintaining diversity, ES avoids premature convergence and is well-suited for QD optimization, which aims to generate diverse high-performing solutions rather than a single global optimum.

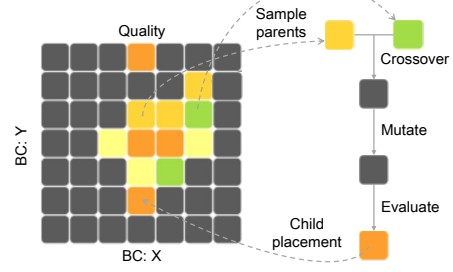

Figure 2: **MAP-Elites flow.**

**MAP-Elites** is a key algorithm within the QD paradigm, designed to optimize for both performance and diversity by maintaining an archive of elite solutions across a discretized space of behavioral characteristics (BCs). Each cell in this archive corresponds to a unique combination of BCs, storing the best solution found for that cell. Figure 2 illustrates an optimization step: two parents are sampled from the archive, and a child solution is generated through crossover and mutation. The child's fitness and BCs are then evaluated to determine its placement in the archive, replacing the current entry if it performs better.

## 3   METHODS

To avoid the laborious tuning of the data mixing ratio and the objective function, we introduce CycleQD. It is built on MAP-Elites, a recognized implementation of QD, but distinguishes itself from traditional MAP-Elites systems in three key ways (see Algorithm 1 for the pseudo-code):

**Alternating Quality and BCs:** Pugh et al. (2016) emphasizes the importance of well-aligned BCs with the quality (i.e., task metric that needs to be optimized). Traditional systems typically set the design at the onset and do not alter them throughout the QD process. In contrast, CycleQD uses task metrics as both quality and BCs, ensuring alignment and dynamically alternating them during the QD process (lines 6-7 in Algorithm 1). This step can be viewed as if the archive is rotated by 90 degrees after each generation before proceeding to the optimization, as is illustrated in Figure 1.

**Model Merging as Crossover:** Unlike existing systems that optimize model parameters directly, CycleQD leverages a model merging algorithm as a crossover operation, replacing the heuristic, hand-designed rules often seen in practice (line 12 in Algorithm 1). I.e., in Figure 2, our crossover operation merges the two parent models. In CycleQD, we fine-tune expert agents, each of which only needs to specialize in one task, and use them as the seed models to initialize the model archives (lines 2-4 in Algorithm 1).

**SVD-based Mutation:** Traditional mutation operations in genetic algorithms like MAP-Elites typically introduce random perturbations from a pre-defined distribution to explore new regions (i.e., the mutation operation in Figure 2). In contrast, CycleQD utilizes perturbations aligned with the principal components of the models' parameters matrices, facilitating exploration outside the convex regions formed by the parent models while avoiding overfitting (line 13 in Algorithm 1).

CycleQD repeats these three steps in each generation, and the archives, originally only occupied by the expert models, accommodate more and more capable and diverse models, as is illustrated by Figure 6 in the Appendix.

### 3.1   ALTERNATING QUALITY AND BEHAVIOR CHARACTERISTICS

CycleQD manages $K$ archives, each dedicated to tracking the LLMs that specialize in one of the $K$ agent skills. Each archive evaluates LLM performance using $K$ tasks, which serve dual roles as both the quality and BCs. An archive is structured as a lattice, the size of which is defined by $\prod_{k \neq i} d_k$, where $d_k$ is the number of bins defined by the $k$-th BC, and the $i$-th BC is excluded from this particular archive as it is treated as the quality metric. In generation $t$, the $i$-th archive is selected for QD computation where $i = t \mod K$ (see Figure 1 for an illustration with $K = 3$). This process

---

**Algorithm 1** CycleQD

---

1: **function** TRAIN($A$, EXPERTS)  ▷ $A$ are the archives, EXPERTS are the expert LLM agents
2:   **for** $i \leftarrow 1$ **to** $K$ **do**  ▷ $K$ is the number of tasks
3:     $A[i] \leftarrow$ UPDATEARCHIVE($A[i]$, EXPERTS)  ▷ Place experts properly in $i$-th archive
4:   **end for**
5:   **for** $t \leftarrow 1$ **to** $N$ **do**  ▷ $N$ is the total number of generations
6:     $i \leftarrow t \mod K$  ▷ Cyclically swaps the task (archive) for the next QD step
7:     $A \leftarrow$ QDSTEP($A, i$)  ▷ Conduct one step of the QD algorithm
8:   **end for**
9: **end function**

10: **function** QDSTEP($A, k$)  ▷ $A_k$ is the archive corresponding to the $k$-th task
11:   $p_1, p_2 \leftarrow$ SAMPLE($A[k]$)  ▷ $p_1$ and $p_2$ are the parents, SAMPLE() is detailed in Sec 3.1
12:   $c \leftarrow$ CROSSOVER($p_1, p_2$)  ▷ $c$ is the child, CROSSOVER() is detailed in Sec 3.2
13:   $c \leftarrow$ MUTATE($c$)  ▷ MUTATE() is detailed in Sec 3.3
14:   **for** $i \leftarrow 1$ **to** $K$ **do**
15:     $A[i] \leftarrow$ UPDATEARCHIVE($A[i], c$)  ▷ Place $c$ properly in $i$-th archive
16:   **end for**
17: **end function**

---

utilizes all available data from the $K$ tasks to evaluate and update the archives without the need for adjusting data ratios or objective functions.

CycleQD shares knowledge across $K$ archives for efficient optimization. Specifically, in generation $t$, a new model is created from the $i$-th archive by sampling, crossover, and mutation. This model is used to update not only the $i$-th archive but all $K$ archives (lines 14-15 in Algorithm 1). This allows for more effective utilization of the new model and helps facilitate optimization across $K$ tasks.

We employ task performance metrics as our BCs, imbuing them with concrete performance implications. This allows for the design of an Elite sampling algorithm aimed to expand the Pareto frontier. Our Elite sampling algorithm is inspired by Wang et al. (2023) which expand the Pareto frontier not only in quality but also within BCs. However, there are two major differences in CycleQD's setup: (1) Both Q and BCs are task metrics and are treated equally in a cyclic process, (2) We aggregate Q and BC values into a single metric to drive the frontier towards higher performance. Specifically, this frontier is defined by the models achieving high performance across these BCs. Concretely, when the $i$-th archive is active, a model $j$ inside it is sampled with probability $P_j = \frac{\gamma_j}{\sum_{n=1}^{N} \gamma_n}$ where $N$ is the number of models in this archive and $\gamma_j$ is calculated as follows:

$$\gamma_j = \prod_{i=1}^{K} \left( \alpha_{\text{low}} + \frac{f_{j,i} - \min(f_{1:N,i})}{\max(f_{1:N,i}) - \min(f_{1:N,i})} (\alpha_{\text{high}} - \alpha_{\text{low}}) \right)$$

Here, $f_{j,i}$ indicates the $j$-th model's performance on the $i$-th task. $(\alpha_{\text{low}}, \alpha_{\text{high}})$ are hyper-parameters for the purpose of normalization. Elite sampling strategically favors models that excel across the quality and various BCs, enhancing the efficiency and evolutionary potential of CycleQD.

## 3.2 MODEL MERGING BASED CROSSOVER

Training an LLM to specialize in a single task does not suffer from problems such as the data ratio tuning, the complex combinations of objective functions, and etc. It is therefore straightforward to initially train a set of single-task experts using conventional fine-tuning methods, and then use model merging algorithms to develop more capable agents. In CycleQD, our model merging crossover operator employs the parameter space merging scheme described in Akiba et al. (2024), which creates a new model by merging task vectors at the model level (Ilharco et al., 2022). Specifically, for a pre-trained base LLM with parameters $\theta_{\text{base}} \in \mathbb{R}^d$ and its fine-tuned LLM with parameters $\theta \in \mathbb{R}^d$, we define a task vector as $\tau = \theta - \theta_{\text{base}}$. The crossover operator then generates a model's parameters by combining these task vectors: $\theta_{\text{child}} = \theta_{\text{base}} + (\omega_1/(\omega_1 + \omega_2))\tau_{p_1} + (\omega_2/(\omega_1 + \omega_2))\tau_{p_2}$, where $\tau_{p_1}$ and $\tau_{p_2}$ are the parents' task vectors. Here, $\omega_1$ and $\omega_2$ are i.i.d samples from $\mathcal{N}(\mu, \sigma^2)$, and $(\mu, \sigma)$ are predetermined hyper-parameters that remain fixed during the experiments. These $\omega$'s do

Figure 3: **SVD-based mutations.** Left: A task vector $\tau$ contains multiple parameter matrices $\tau_l$. E.g., the query projection matrix from the attention block in layer 1, the key projection matrix from the attention block in layer 2, etc. Those that have a rank $r > 1$ can be decomposed using SVD into $r$ components. Right: SVD-based mutation samples a vector $w \in \mathbb{R}^r$ and scales each component by $w_i$, essentially adding perturbations that are aligned with the "directions" of the components.

not need to be positive numbers. By allowing negative component weights, the merged model has more freedom in optimizing its task vectors, the process of which is automatically carried out via evolutionary search. We normalize $(\tau_{p_1}, \tau_{p_2})$'s mixing coefficients to ensure the merged weights do not become outliers and cause problems for the downstream layers.

Additionally, there is potential to enhance the model merging process by training a neural network policy $(\mu, \sigma) \leftarrow \pi_\phi\big(\{bc_1, ..., bc_k\}_{p_1}, \{bc_1, ..., bc_k\}_{p_2}\big)$ that learns the optimal distribution parameters conditioned on the BC grid IDs $\{bc_1, ..., bc_k\}$ of the parents ($k - 1$ in total for each parent because one serves as the quality). However, while this approach could improve the efficiency of model merging, it also introduces a greater initial learning burden in CycleQD.

### 3.3 SVD-BASED MUTATION

The formulation of our model merging based crossover bears one obvious limitation: The construction of $\theta_{\text{child}}$ is a linear combination of $(\tau_{p_1}, \tau_{p_2})$, which are themselves linear combination of their parents. The entire process then reduces to finding the optimal linear combination of the expert models' parameters. Due to this, $\theta_{\text{child}}$ will likely be trapped in the "convex region" in the performance space formed by the expert models, yet extrapolation that leads to improved performance is desired. We introduce a mutation function $\theta_{\text{child}} = h(\theta_{\text{child}})$ after the crossover to get rid of this limitation.

In conventional methods, the mutation function $h(\theta_{\text{child}})$ is often an operator that samples perturbations from a pre-defined distribution (e.g., Gaussian with pre-determined mean and covariance matrix) and add them to the gene being mutated. We find this setting introduce excess freedom and lead to overfitting in the final model. Instead, we propose to sample perturbations along the "directions" of the components from the model's parameter matrices. This mutation operation is mathematically defined as: $h(\theta_{\text{child}}) = \theta_{\text{base}} + \text{concat}\big([U_l(\Sigma_l w)V_l^\mathsf{T}]_{l=1}^L\big)$, where $U_l \in \mathbb{R}^{m \times r}$, $\Sigma_l \in \mathbb{R}^{r \times r}$ and $V_l \in \mathbb{R}^{n \times r}$ are the SVD components of $\tau_l = U_l \Sigma_l V_l^\mathsf{T}$, the $l$-th parameter matrix in the task vector $\tau_{\text{child}} = \theta_{\text{child}} - \theta_{\text{base}}$ (i.e., $\tau_l$ is a reshaped subarray of $\tau_{\text{child}}$). The perturbation vector $w \in \mathbb{R}^r$ is sampled from a uniform distribution of boundaries $[0, w_{\max}]$, where $w_{\max} \in \mathbb{R}^+$ is a hyper-parameter. Our SVD-based mutation becomes a pass-through operation for parameter matrices whose rank is one (e.g., those belonging to layer-normalization layers).

This approach provides two key advantages. First, SVD focuses mutations on the fundamental building blocks of the task vector, which represent the essential components of agent skills. By constraining perturbations to these meaningful directions, we avoid the excessive exploration inherent in random perturbations, reducing the risk of overfitting. Second, the ability to manipulate individual components (i.e., $\omega$) offers finer control over $\theta_{\texttt{child}}$, enabling targeted modifications at a granular level. Figure 3 illustrates this process.

### 3.4 MODEL AGGREGATION

CycleQD maintains $K$ archives during the optimization process, LLMs in each of these archives are trained to acquire one of the $K$ agent skills. In the end, CycleQD returns hundreds or even thousands of LLM agents with various specialties and characteristics. These LLM-based agents are versatile and can facilitate multi-agent research in a diverse set of directions (we discuss this topic in Section 6).

On the other hand, our goal in this work is to create a single model that achieves high performance across multiple tasks, and a model aggregation method is therefore necessary. Our model aggrega-

tion algorithm is straightforward, and can be summarized as: $\theta_{\text{agg}} = \theta_{\text{base}} + \sum_{k=1}^{K} \beta_k \tau_k$, where $\tau_k$ is the task vector from the elite model in the $k$-th archive, with ties resolved by selecting the most recent model. $\beta_k = \exp(f_k) / \sum_{i=1}^{K} \exp(f_i)$ (i.e., softmax function) is the mixing coefficient calculated from the elite model's task performance $f_k$ in the $k$-th archive. In this paper, all experimental results from CycleQD are produced with this aggregated model $\theta_{\text{agg}}$.

# 4 EXPERIMENTS

## 4.1 EVALUATION ON COMPUTER SCIENCE TASKS

### 4.1.1 TASK DESCRIPTION

In this experiment, our goal is to train LLMs to master three agent skills in computer science: coding, Operation System (OS) manipulation and Database (DB) query generation. We adopt the MBPP+ (Mostly Basic Python Programming) dataset from EvalPlus (Liu et al., 2023a) for coding skill development, and utilize the OS and DB datasets from AgentBench (Liu et al., 2024) for training. For the coding task, we optimize and report the pass@1 metric, whereas in OS and DB tasks we use the success rate. Please refer to Appendix A.1.1 for extra and detailed task related setups.

### 4.1.2 CYCLEQD SETUPS

**Experts:** We employ LLAMA-3-8B-INSTRUCT MODEL (Dubey et al., 2024) as our base model, and use supervised fine-tuning to create LLM experts in coding, OS and DB. For the OS and DB experts, we use the OS and DB training datasets from Agent-FLAN Chen et al. (2024). The coding expert is fine-tuned on a combination of the MAGICODER-EVOL-INSTRUCT-110K and MAGICODER-OSS-INSTRUCT-75K datasets (Wei et al., 2024). See detailed configuration in Appendix A.1.2

**Datasets:** We use the MBPP+ dataset, as well as the test and development datasets from OS and DB. To ensure that the experts have the same task metrics across tasks, each dataset is split evenly into training and test splits. For OS, problems that could not be solved by either the expert models and GPT models are excluded beforehand to reduce computation cost.

**Hyper-parameters:** We use the performances of the three experts to determine the lower and upper bounds of the BC dimension. Specifically, the lower bound is set at 85% of the performance achieved by the least proficient expert, while the upper bound is set at 115% of the performance achieved by the most proficient expert. All BCs are then evenly divided into 15 bins between the lower and upper bounds. We limit the number of models in each bin to one, and run CycleQD for 1200 generations. Since the quality and BCs are alternated in each generation, this is equivalent to optimizing for the three skills for 400 generations each. See more detail in Appendix A.1.3

### 4.1.3 BASELINES

Our baselines can be categorized into fine-tuning based and merging based models.

**Fine-tuning Based Models:** These are models 3-6 in Table 1. In addition to the expert LLMs mentioned earlier, we also combine all the data from the three tasks (including the extra MAGICODER-EVOL-INSTRUCT-110K and MAGICODER-OSS-INSTRUCT-75K datasets) and continue training the LLAMA3-8B-INSTRUCT base model with supervised fine-tuning to develop an extra baseline (model 6 in Table 1). In this baseline, we don't tune the data ratio and use cross-entropy (i.e., next token prediction) as our objective function.

**Merging Based Models:** These are models 7-10 in Table 1. We first introduce a naive merging method where the merged model is produced by taking an average of the three experts' task vectors (model 7). Similar to this baseline, we introduce three additional learning based model merging methods where, instead of taking an average, the coefficients of a linear combination of these task vectors is learned through gradient descent on policy gradients (model 8), CMA-ES, an evolutionary algorithm on the raw rewards (model 9), and NSGA-II (Deb et al., 2002), another evolutionary algorithm that optimizes multiple objectives simultaneously (model 10).

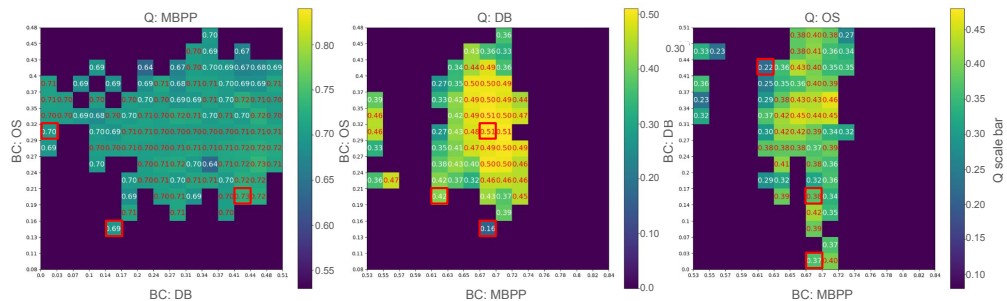

Figure 4: **CycleQD Archives from Computer Science Tasks.** In each archive, the two axes show the BCs, and the color intensities indicate the LLM agent's quality in that grid. These archives are obtained after 1200 generations of CycleQD. The red bounding boxes indicate the grids where expert models were present. See Appendix A.1.4 for archive development across generations.

### 4.1.4 RESULTS

**Overall:** We summarize our results from this experiment in Table 1. In addition to the baselines, we also include the base model and the GPT models for references (highlighted with a lightgray background). The first thing to notice is that each expert model (models 3-5), fine-tuned specifically for its assigned task, performs better than the LLAMA3-8B-INSTRUCT base model (model 2), laying the stepping stones for CycleQD and other model merging based methods. After evolutionary computing, our method, averaged across the three tasks, outperforms all baselines, and is approaching the performance of GPT-3.5-TURBO. Specifically, CycleQD achieves the highest scores on the coding and the OS tasks among the baselines, and has only a mild performance drop on the DB tasks compared with the expert. Figure 4 gives the archives in CycleQD at the end of the experiment. It is easy to see that CycleQD has managed to "illuminate" the three archives.

**Comparison with Fine-tune Based Methods:** A notable result from the table is that, the model fine-tuned on all the datasets (model 6) is only partially and marginally better than the three experts, despite being trained on a larger dataset. On the other hand, the expert models can be easily trained because they do not require the data ratio or object function adjustment. The drawback of these experts is that they specialize in only one task, prohibiting their usage in wider scenarios. This result proves the difficulty with conventional methods, underscoring the importance and technical contributions of CycleQD.

**Comparison with Merging Based Methods:** CycleQD also outperforms conventional model merging based methods (models 7-10), and we conjecture that this is due to three reasons. First, these methods lack a mutation operation, which we believe is crucial for enabling the merged model to es-

Table 1: **Evaluation on computer science tasks.** We report pass@1 for MBPP and success rates for the DB and OS tasks. The base model and the GPT models (in gray) are included for completeness. The MBPP results for the GPT models are extracted from Dubey et al. (2024).

| # | Methods | MBPP | DB | OS | Avg |
|---|---------|------|-----|-----|-----|
| 0 | GPT-4 | 83.6 | 36.5 | 63.7 | 61.3 |
| 1 | GPT-3.5-TURBO | 82.0 | 41.6 | 38.5 | 53.7 |
| 2 | Llama3-8B-Instruct (base model) | 67.3 | 5.3 | 25.2 | 32.6 |
| | *Fine-tuning Based Methods* | | | | |
| 3 | Fine-tuning (Coding expert) | 70.4 | 21.2 | 20.7 | 37.4 |
| 4 | Fine-tuning (DB expert) | 65.8 | **42.4** | 28.5 | 45.6 |
| 5 | Fine-tuning (OS expert) | 66.3 | 0.0 | 30.4 | 32.2 |
| 6 | Fine-tuning (All) | 67.3 | 37.1 | 36.7 | 47.0 |
| | *Merging Based Methods* | | | | |
| 7 | Merging (w/o learning) | 72.9 | 24.7 | **42.6** | 46.7 |
| 8 | Merging (learning w/ GD) | 69.3 | 41.2 | 29.6 | 46.7 |
| 9 | Merging (learning w/ CMA-ES) | 69.3 | 41.2 | 30.2 | 46.9 |
| 10 | Merging (learning w/ NSGA-II) | 75.9 | **42.4** | 36.4 | 51.6 |
| 11 | CycleQD (Ours) | **76.4** | 38.2 | **42.6** | **52.4** |

Table 2: **Ablation studies.** We add and highlight a different treatment from the previous trials.

| # | Trials | MBPP | DB | OS | Avg |
|---|--------|------|-----|-----|-----|
| 0 | QD + No mutation + Random sampling | 70.4 | 28.8 | **43.7** | 47.6 |
| 1 | CycleQD + No mutation + Random sampling | 72.9 | 33.5 | 41.9 | 49.4 |
| 2 | CycleQD + Gaussian mutation + Random sampling | 73.4 | 30.0 | 42.2 | 48.5 |
| 3 | CycleQD + SVD mutation + Random sampling | 75.9 | **38.2** | 41.1 | 51.7 |
| 4 | CycleQD + SVD mutation + Elite sampling | **76.4** | **38.2** | 42.6 | **52.4** |

cape the "convex region" created by the seed models. For example, model 9 scores lower on all three tasks compared to the experts. In contrast, CycleQD significantly surpasses expert performance in coding and OS tasks. Second, CycleQD's cyclic optimization mechanism incorporates elements resembling non-dominant sorting in NSGA-II (model 10). However, unlike the latter, which handles all tasks simultaneously and relies on crowding distance for diversity, CycleQD alternates between tasks and naturally encourages diversity via the QD mechanism. This allows CycleQD to be more effective. Lastly, we suspect that the absence of a mechanism to rehabilitate temporarily underperforming models contributes to this discrepancy. Given the lengthy computation pipelines in LLMs, which include numerous layers, even a single malfunctioning layer can degrade overall performance. Unlike QD, the baselines do not have a way to recover these layers, potentially promising models are prematurely discarded.

### 4.1.5 Ablation studies

To get insights into the design choices in CycleQD and show how they contribute its effectiveness, we conduct a series of ablation studies and summarize the results in Table 2.

**CycleQD vs QD:** We compare conventional QD with CycleQD in trials 0 and 1. Specifically, we run QD for three runs. Within each run, one of the task's metric is treated as the quality while the others are regarded as the BCs. We control each trial to have an equal computing budget as for CycleQD, and all other hyper-parameters are identical to CycleQD. We use the same model aggregation method to merge the best models from these three runs (see Section 3.4). The better performance from CycleQD suggests the importance of alternating the quality and BCs.

**Mutation Methods:** We focus on the impact of different mutation operations with trials 1-3. Although we mentioned earlier that mutation helps the merged model extrapolate in the performance space, naively designed mutations give excess freedom and can lead to overfitting. This is what happens to trial 2, the performance of which is even worse than trial 1 that does not have any mutation operations. On the other hand, our SVD-based mutation (trial 3) successfully avoids the problem and delivers a much better performance.

**Sampling Methods:** Finally, we demonstrate the importance of Elite sampling in CycleQD in trial 4, which has the identical settings as model 11 in Table 1, and gives the best score in the ablation studies. Elite sampling is effective because it prioritizes merging high-performing models, the result of which helps expand the Pareto frontier in each archive more effectively.

In summary, our ablation studies validate the importance of our design choices and shows that improvements aren't merely additive - poor design choices can harm performance while our well-designed components work synergistically (e.g., comparing rows 1 and 2 in Table 2). The cumulative improvement from baseline QD to our final CycleQD is substantial at 4.8 percentage points (from 47.6% to 52.4%). This is a significant improvement in the context of LLMs performing complex agent tasks, particularly considering our model approaches GPT-3.5-TURBO's performance despite having significantly fewer parameters.

### 4.1.6 Skill generalization and language capabilities

In addition to agent skills, we evaluate our model on out-of-distribution coding tasks and language understanding across six categories, please see Appendix A.1.6 for detailed task descriptions. Table 3 shows the results across these tasks, where we normalize the scores against the LLAMA3-8B-INSTRUCT base model. On average, CycleQD outperforms the three fine-tuned experts, providing better overall generalization across tasks. Our method generalizes well on the coding tasks while re-

Table 3: **Generalization performance.** The scores are normalized against the base model. CycleQD generalizes well to other coding tasks unseen in the training while retaining its language capabilities.

| Model | Coding Tasks | | Language Tasks | | | | Avg |
|---|---|---|---|---|---|---|---|
| | HUMANEVAL+ | BigCodeBench | Reasoning | GSM8K | RC | CommonSense | |
| MBPP expert | 1.18 | 0.97 | 0.57 | 0.82 | 0.94 | 1.03 | 0.92 |
| DB expert | 0.80 | 0.84 | 0.84 | 0.87 | 0.98 | 0.98 | 0.89 |
| OS expert | 0.94 | 0.90 | 0.98 | 0.93 | 0.99 | 0.99 | 0.95 |
| CycleQD | 1.10 | 1.03 | 0.95 | 0.88 | 0.98 | 1.02 | 0.99 |

Table 4: **Merging image segmentation Models**. The scores are normalized against the performance of the corresponding expert models. Model similarity shows how close the pair of expert models are (the larger the similarity values, the closer the models are).

| # | Expert A | Expert B | Score A | Score B | Avg Score | Model Similarity |
|---|---|---|---|---|---|---|
| 0 | CAM | POL | 0.95 | 0.99 | 0.97 | 0.98 |
| 1 | CAM | SKL | 0.85 | 0.99 | 0.92 | 0.97 |
| 2 | CAM | LEA | 0.51 | 0.89 | 0.70 | 0.88 |
| 3 | POL | SKL | 0.98 | 0.95 | 0.96 | 0.99 |
| 4 | POL | LEA | 0.40 | 0.84 | 0.62 | 0.93 |
| 5 | SKL | LEA | 0.83 | 0.84 | 0.83 | 0.95 |

training performance on language tasks. In contrast, the MBPP expert, while delivering competitive scores on the coding tasks, shows significantly lower performance on specific tasks in the Reasoning category. This sharp decline in performance suggests the occurrence of catastrophic forgetting during fine-tuning, where the model loses its ability to generalize across tasks outside its specialization. These findings once again underscore the efficacy of CycleQD in achieving superior and more consistent performance across diverse tasks compared to traditional fine-tuning approaches.

## 4.2 Evaluation on image segmentation Tasks

### 4.2.1 Task description

Besides to its applications for LLMs, CycleQD serves as a versatile method for integrating expert models across various data modalities beyond text. For example, we include a vision question answering (VQA) task in addition to the CS tasks and find CycleQD to be able to outperform the experts (see Section A.1.5). In this experiment, we go further beyond and extend CycleQD to the fusion of multiple Segment Anything Models (SAM), which are state-of-the-art computer vision models designed for image segmentation tasks. Specifically, our objective is to merge pairs of SAM models, A and B, to create models whose capabilities encompass the skill sets of both A and B.

### 4.2.2 CycleQD setups

**Experts:** We select four tasks within the image segmentation domain, each supported by extensive datasets, for training specialized models: Camouflaged Object Segmentation (CAM), Polyp Segmentation (POL), Skin Lesion Segmentation (SKL), and Leaf Segmentation (LEA). CAM detects objects in cluttered environments, making segmentation more challenging than standard tasks. POL identifies polyps in endoscopic images, essential for early colorectal cancer detection. SKL detects various skin lesions in medical images. LEA identifies plant leaves in agricultural images, supporting disease control and improving crop quality. We employ SAM-ViT Huge model as our base model, which we fine-tune to develop these experts. See Appendix A.2.1 for details on the datasets used for fine-tuning and CycleQD setups.

**Hyper-parameters:** CycleQD's hyper-parameters remain the same as in Section 4.1.2.

### 4.2.3 Results

**Overall:** Table 4 shows the performance of the models merged by CycleQD, where scores are normalized against the expert models. You can see visualization results in Appendix A.2.2. In general, CycleQD is able to merge the experts successfully, with top models retaining more than

90% of the experts' performance (e.g., models 0, 1, and 3). On the other hand, models 2, 4 and 5 are less successful. This leads us to conduct further analysis and we report the findings next.

**Analysis:** We report the similarities between experts A and B in the last column in Table 4. A strong correlation of 0.83 between the averaged scores and these similarities indicate both the limitations and potential areas for enhancement in CycleQD. We define the similarity between models A and B as the average cosine similarity of the singular value vectors (derived from the task vectors) across all layers in both models: $s = (1/L) \sum_{i=1}^{L} \cos\big(\text{diag}(\Sigma_{i,A}), \text{diag}(\Sigma_{i,B})\big)$. Here, $L$ is the number of weight matrices with a rank greater than 1, and $\Sigma_{i,*}$ denotes a diagonal matrix of singular values from the $i$-th weight matrix in the task vector. Although this summarizing metric does not fully encapsulate the models' characteristics, CycleQD tends to perform well when the models exhibit high similarity. This observation underpins our approach of incorporating this similarity metric as a regularization technique during model fine-tuning. Notably, a similar metric has been employed as the "proximal term" in optimizing heterogeneous networks within the realm of federated learning, as evidenced by research documented in (Li et al., 2020). This precedent lends credence to our strategy, suggesting it is grounded in established methodologies.

## 5 RELATED WORKS

LLM skill acquisition research has expanded beyond conversational tasks, focusing on enabling autonomous problem-solving through the LLM-as-Agent paradigm (Xi et al., 2023; Wang et al., 2024b). Techniques like ReAct (Yao et al., 2023) integrate reasoning with action, but skill acquisition through training remains limited, often relying on standard fine-tuning (Zeng et al., 2024; Qin et al., 2024), which struggles with balancing multiple skills. QD (Pugh et al., 2016) is an evolutionary paradigm aimed at discovering diverse, high-performing solutions. Algorithms like MAP-Elites (Mouret & Clune, 2015) and MOQD (Pierrot et al., 2022) have refined QD for tasks with diverse behaviors and conflicting objectives. While QD has been applied to neural networks and LLM prompts (Guo et al., 2024; Xue et al., 2024), evolving LLM weights remain unexplored. Model merging, which combines capabilities across pre-trained models at lower training costs, has advanced through techniques like sparsification (Yadav et al., 2023; Stoica et al., 2024) and optimization (Yang et al., 2024). Our work integrates QD with model merging, allowing LLM components to be preserved and optimized for diverse tasks. Please refer to Section B for more related works.

## 6 CONCLUSIONS AND FUTURE WORKS

In this work, we introduced CycleQD, a compelling modification of the conventional LLM fine-tuning pipeline, integrating evolutionary algorithms for agent skill acquisition. CycleQD begins with single-task experts and utilizes QD to continuously generate hundreds of LLM-based agents, possessing diverse characteristics and exhibiting a broader range of skills than the initial experts. Through a dynamic process that cyclically swaps quality and BCs, coupled with a model merging crossover operation and an SVD-based mutation operator, CycleQD has successfully enabled LLMs to master computer science skills. Our method not only outperforms baseline approaches but also achieves on-par performance with GPT-3.5-TURBO, while generalizing to out-of-distribution tasks and retaining the language capabilities. Furthermore, CycleQD's applicability extends across domains to image segmentation models, demonstrating its broad utility.

In terms of limitations, we acknowledge that the success of model merging hinges on the compatibility of the source models. CycleQD may encounter challenges when the expert models originate from highly divergent settings. One way to address this is incorporating model similarity as a regularization term during the expert training. Furthermore, this research represents an initial foray into the integration of QD and evolutionary model merging for agent skill acquisition. There is substantial potential for improvement by leveraging advanced methodologies from both fields, such as integrating strategies from CMA-ME (Fontaine et al., 2020) and PGA-MAP-Elites (Nilsson & Cully, 2021) to enhance the efficiency of the learning process. Looking ahead, a promising direction for future research lies in multi-agent systems. Considering CycleQD generates an archive of diverse agents, orchestrating these agents to collaborate and compete opens up exciting possibilities for scientific exploration and practical applications.

AUTHOR CONTRIBUTIONS

Yujin Tang and Takuya Akiba initiated the project. So Kuroki co-designed the algorithm, implemented CycleQD, conducted the experiments and the studies. Taishi Nakamura developed the agentic tasks, the expert models and the baseline methods. Takuya Akiba implemented the parallel distributed optimization and proposed the SAM experiment. Yujin Tang proposed the algorithm, designed and helped with the experiments, and managed the project. All authors contributed to the writing and approved the final version.

ACKNOWLEDGEMENTS

The authors would like to thank Robert Lange for providing valuable discussions and feedback while drafting the text, and Kou Misaki and Qi Sun for providing infrastructure support. This paper is based on results obtained from a project, JPNP20017, subsidized by the New Energy and Industrial Technology Development Organization (NEDO).

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

## A APPENDIX

### A.1 COMPUTER SCIENCE TASKS

#### A.1.1 EXTRA TASK SETUPS

The MBPP+ dataset, based on the original MBPP, uses a subset of 399 hand-verified problems from MBPP-sanitized to ensure well-formed programming tasks. Each problem includes a problem statement, function signature, and test cases. EvalPlus extends this dataset with additional test cases to provide a more rigorous evaluation of code generation capabilities. For our evaluation, we use the pass@1 metric on the Base Tests of MBPP+, which reflects the model's ability to generate correct code on the first attempt using the original test cases. The OS dataset evaluates LLMs in genuine interactive bash environments (Ubuntu Docker) on human questions with deterministic answers and practical operational tasks. The DB dataset assesses LLMs' abilities to operate on real databases via SQL, encompassing the complete pipeline of database analysis with authentic SQL interfaces and multiple tables. Both OS and DB datasets use success rate as the primary evaluation metric. The OS tasks are designed to be solved within a maximum of 5 interaction turns and employ a 1-shot setup. The DB tasks have a similar interaction limit but are evaluated in a 0-shot manner.

#### A.1.2 TRAINING CONFIGURATION FOR EXPERTS

We utilize llm-recipes (Fujii et al., 2024) (commit 606cdfb) for fine-tuning. We adopt the AdamW optimizer (Loshchilov & Hutter, 2019) with $\beta_1 = 0.9$ and $\beta_2 = 0.95$. A global batch size of 64 is used across all fine-tuning processes. We employ cosine learning rate scheduling with a range of $[4 \times 10^{-6}, 2 \times 10^{-5}]$, starting with a linear warmup for the first 10% of the total training steps. The OS and DB experts are trained for 1 epoch, while the code model is trained for 3 epochs due to its larger training data size.

#### A.1.3 CYCLEQD HYPER-PARAMETERS

We set $\alpha_{\text{low}} = 0.5$ and $\alpha_{\text{high}} = 0.8$ in Elite sampling, $\mu = 1.0$ and $\sigma = 0.03$ in model merging-based crossover, and $w_{\text{max}} = 0.3$ in our SVD-based mutations. These hyperparameters are determined in preliminary studies, where we arbitrarily determined a set of values, ran CycleQD for a small number of generations and picked the set with the best performance. We believe the performance of CycleQD can be further improved when incorporating sophisticated hyperparameter searching methods.

To investigate the influence of the number of generations on CycleQD performance, we analyzed the average scores of the computer science tasks every 100 generations. Figure 5 illustrates these results. The performance gradu-

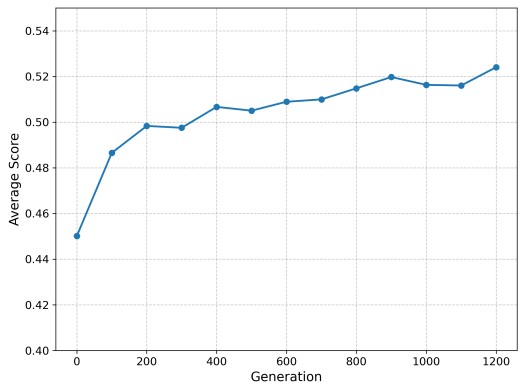

Figure 5: **Averaged performance on computer science tasks.** Evaluated every 100 generations.

ally increases over the generations with no obviously significant oscillations. While still showing a slight upward trend beyond this point, it stabilizes around 1000 generations.

#### A.1.4 CYCLEQD DEVELOPMENT OF ARCHIVES ACROSS GENERATIONS.

Figure 6 shows the development of archives across generations. The archives are shown in increments of 300 generations from top to bottom. The corresponding generation for each archive is displayed on the left side of the figure. The red bounding boxes indicate the grids where expert policies were present in each archive. It can be observed that the frontier of the archives expands with each passing generation.

### A.1.5 ADDING A VISION QUESTION ANSWERING TASK

In addition to the computer science tasks, we add a visual question answering (VQA) task to Cy-cleQD to further demonstrate its generally applicability across tasks that differ drastically. The VQA task lies within the vision-language modeling (VLM) domain, which presents challenges when merging it with tasks from LLM domain.

Specifically, we use the TextVQA dataset (Singh et al., 2019) as part of the CycleQD experiments along with the coding, DB, and OS tasks. The experimental setup follows the same configurations described in Section 4.2.2, including parameters and the 1200 generations training budget. To create a VQA expert, we optimize the LLAMA3-LLAVA-NEXT-8B model using 4000 samples from the TextVQA data as training data. Additionally, we allocate 500 distinct samples, separated from the training data, for CycleQD training. These 500 samples are evenly divided into optimization and test datasets.

During the CycleQD optimization process, we extract the LLAMA3-8B-INSTRUCT component of the VLM and perform merge and mutate operations. The results, presented in Table 5, show that CycleQD achieves the best average performance compared to both the base and the expert models. Notably, CycleQD outperforms the several expert models on VQA, coding, and OS tasks, highlighting its effectiveness in combining knowledge across diverse domains.

Table 5: **Evaluation on computer science tasks and VQA.** We add VQA task to computer science tasks and evaluate CycleQD.

| # | Methods | VQA | MBPP | DB | OS | Avg |
|---|---|---|---|---|---|---|
| 0 | Llama3-8B-Instruct (base model) | 39.0 | 67.3 | 5.3 | 25.2 | 34.2 |
| | *Experts models* | | | | | |
| 1 | VQA expert | 51.4 | 4.0 | 0.0 | 1.5 | 14.2 |
| 2 | Coding expert | 32.9 | 70.4 | 21.2 | 20.7 | 36.3 |
| 3 | DB expert | 45.4 | 65.8 | **42.4** | 28.5 | 45.5 |
| 4 | OS expert | 46.1 | 66.3 | 0.0 | 30.4 | 35.7 |
| 5 | CycleQD (Ours) | **54.1** | **72.9** | 32.4 | **39.6** | **49.7** |

### A.1.6 BENCHMARK DATASETS FOR SKILL GENERALIZATION AND LANGUAGE UNDERSTANDING

In the Coding category, we employ two key benchmarks: HUMANEVAL+ (Liu et al., 2023a), eval-uated in a 0-shot setting using the pass@1 metric on the Base Tests, and BigCodeBench (Zhuo et al., 2024), which is designed to assess code generation with diverse function calls and complex instruc-tions. We report the average Pass@1 scores for both BigCodeBench-Complete and BigCodeBench-Instruct variants in the full setting, using 0-shot evaluation. For the General Knowledge and Rea-soning (Reasoning) category, we utilize two comprehensive benchmarks: MMLU (Hendrycks et al., 2021) with a 5-shot evaluation, and BBH (Big Bench Hard) (Suzgun et al., 2023) using a 3-shot chain-of-thought (Wei et al., 2022) prompting approach. These benchmarks provide a holistic as-sessment of our model's ability to handle a wide range of knowledge-based and reasoning tasks. To evaluate Mathematical Reasoning, we employ the GSM8K benchmark (Cobbe et al., 2021) with a 4-shot evaluation, challenging our model's ability to solve complex mathematical problems. For Read-ing Comprehension (RC), we use two established benchmarks: SQuAD2 (Rajpurkar et al., 2018) and TriviaQA (Joshi et al., 2017), both evaluated using a 4-shot approach. These datasets assess the model's capacity to understand and reason over complex textual information. In the Commonsense Reasoning (CommonSense) category, we employ three diverse benchmarks: HellaSwag (Zellers et al., 2019) for commonsense inference, OpenBookQA (Mihaylov et al., 2018) for elementary-level science question answering that requires both core scientific knowledge and broader common knowledge, and XWinograd (English version) (Tikhonov & Ryabinin, 2021) for cross-lingual com-monsense reasoning and coreference resolution. All these evaluations are conducted with 4-shot prompts.

### A.1.7 Training time comparison

We used NVIDIA H100 GPUs for our experiments. The gradient fine-tuning method (model #6 in Table 1) took approximately 200 GPU hours, and our method (model #11 in Table 1) took about 410 GPU hours (excluding the expert models training time). We wish to point out that (1) the extra time spent on CycleQD is mostly due to agentic task evaluations rather than the optimization, (2) due to GPU memory constraints, the merging process was carried out on CPU, (3) CycleQD produces an archive of models whereas fine-tuning generates only one. Furthermore, it is important to note that fine-tuning is a mature and well-established approach with software implementations that are optimized for efficiency. By contrast, the proposed method was developed primarily to assess its performance, and as such, there is ample room for further efficiency improvements.

### A.2 Image segmentation tasks

### A.2.1 Datasets for fine-tuning experts and CycleQD training

We followed Zhong et al. (2024) to prepare the datasets. For Camouflaged Object Segmentation, we use three datasets: COD10K (Fan et al., 2020a), CHAMELEON (Skurowski et al., 2018), and CAMO (Le et al., 2019). Following Fan et al. (2020a) we train on a combined dataset consisting of the 4040 training images from COD10K and CAMO for 20 epochs, randomly splitting 10% of the images from the training set for validation. The model is then tested on the 250 COMO test images. For Polyp Segmentation, we use two datasets: Kvasir (Jha et al., 2019) and CVC-ClinicDB/CVC-612 (Bernal et al., 2015). Following Fan et al. (2020b), we divide the images into a 9:1 ratio for training and testing, resulting in 1450 training images. We then randomly split 20 % of the training set for validation. The model is trained for 30 epochs and tested on the 101 Kvasir test images. For Skin Lesion Segmentation, we use the ISIC 2017 dataset (Codella et al., 2018). We train the model on the 2000 training and 150 validation images for 30 epochs and evaluate it on the 600 test images. For Leaf Segmentation, we use the Leaf Disease Segmentation Dataset (Rath, 2023). We train the model on the 498 training image, using 80% for training and 20% for validation for 30 epochs, and evaluate it on 90 test images.

### A.2.2 Visualization result

The visualization results are shown in Figure 7 for SKL and POL, and Figure 8 for SKL and LEA. In both figures, "Ours" refers to a single merged model via CycleQD, identical to model #3 (for SKL and POL) and model #5 (for SKL and LEA) in Table 4. In Figure 7, our model retains near expert-level performance for both tasks after merging. However, in Figure 8, while our model generally performs at an expert level for both tasks, a slight drop is observed. Notably, in the bottom images of SKL, the model seems to detect the edges of the skin lesions instead of the lesions themselves, as if confusing the task with leaf disease detection. The expert model similarities are high—0.99 for SKL and POL, and 0.95 for SKL and LEA. This corresponds with the observation that the merged model from the highly similar SKL and POL performs better than the one from SKL and LEA.

## B More related works

**LLM Skill Acquisition:** Beyond conversational tasks, increasing attention has been given to enabling LLMs to acquire *skills* to take action. This allows LLMs to function as agents capable of solving problems autonomously, and this approach is known as *LLM-as-Agent* (Xi et al., 2023; Wang et al., 2024b). Benchmarks for LLM-as-Agent have been developed, and research has demonstrated that LLMs can successfully perform tasks such as computer operations and web browsing to a certain extent (Gur et al., 2023; Liu et al., 2023b; Xi et al., 2024; Xu et al., 2024; Tan et al., 2024; Ye et al., 2024). *ReAct* (Yao et al., 2023) is the most frequently used approach for constructing LLM-as-Agent systems, which integrates CoT reasoning with action. ReAct is typically employed through prompting and few-shot in-context learning, where models learn skills from a small number of examples. However, attempts to equip LLMs with such skills through actual training with more examples are still quite limited, with a few studies focusing only on standard fine-tuning (Zeng et al., 2024; Qin et al., 2024). This fine-tuning approach often struggles with balancing multiple agent skills and suffers from the gap between next token prediction loss and actual task performance. Our proposed method uses QD techniques to optimize each skill independently and directly.

**Quality Diversity:** QD is an emerging paradigm in evolutionary computation that aims to discover a diverse collection of high-performing solutions rather than a single optimal result (Pugh et al., 2016). QD algorithms balance two key aspects: *quality*, which represents a solution's performance, and *behavior characterization* (BC), which describes its observable properties. These algorithms maintain an *archive* of diverse, high-quality solutions, continuously updated through an evolutionary process. MAP-Elites (Mouret & Clune, 2015) is a prominent QD algorithm that maintains an archive defined by the BCs, discretizes it into tessellation, and stores the best-performing solutions in each cell to preserve diversity. In each generation, it samples parents from the archive, creates offspring through crossover and mutation, and evaluates their quality and BCs. Offspring replace existing elites if they perform better in their respective cells. This process simultaneously explores diverse behaviors and optimizes performance within each niche, resulting in a map of high-quality, diverse solutions. Recent improvements to MAP-Elites include a new selection method for better diversity (Wang et al., 2023) and an approach that removes the need for predefined niches (Wickman et al., 2023). Additionally, to address potential stagnation in a fixed BC space, Usui et al. (2023) propose dynamic switching of the BC space. CycleQD goes further by aiming for MOQD, rotating both the BCs and the quality across tasks. MOQD (Pierrot et al., 2022) has been proposed to address problems with multiple, potentially conflicting objectives while maintaining diversity. Instead of the fitness, it maximizes a Pareto front hyper-volume in each niche of the descriptor space. CycleQD is a simplified form of MOQD in the sense that it uses task metrics as both BCs and the quality, the product of which approximates the hyper-volume, and the coordinate-ascent style of optimization allows it to avoid complex hyper-volume calculations for higher-dimensional archives.

**Quality Diversity for LLMs:** Evolutionary computation, particularly QD, has occasionally been applied to neural networks and LLMs (Guo et al., 2024; Xue et al., 2024; Akiba et al., 2024). However, there has been no prior research that evolves the main weights of LLMs using QD. Samvelyan et al. (2024) proposed Rainbow Teaming, which applies QD to evolve diverse adversarial prompts for evaluating LLM safety. Bradley et al. (2024) introduced QDAIF, which applies QD with LLM feedback to evolve populations of generated creative texts. Our work represents the first attempt to merge LLMs through QD. We believe this approach holds great promise, as it allows individual components of the model to be preserved even when they are temporarily sub-optimal, with the potential to contribute to future performance improvements. Given the cost and complexity of optimizing the many layers of LLMs, QD offers a unique advantage by retaining useful configurations in archives that may later enhance overall model performance.

**Model Merging:** Model merging refers to combining multiple pre-trained models into a single model. This technique is gaining attention because it can enhance model performance and integrate multiple capabilities at a much lower training cost. Additionally, unlike ensemble methods, model merging does not increase inference costs either. This technique is particularly effective when applied to different fine-tuned versions of the same base model. The most basic method involves a linear combination of weights (Wortsman et al., 2022; Ilharco et al., 2023). More advanced merging techniques have been proposed, incorporating methods such as election and sparsification (Yadav et al., 2023; Yu et al., 2024; Wang et al., 2024a; Stoica et al., 2024). Furthermore, the applicability and performance of these methods have been significantly improved through optimization (Akiba et al., 2024; Yang et al., 2024). Inspired by these merging strategies, CycleQD combines QD with model merging, allowing for more effective synthesis of multiple capabilities in LLMs.

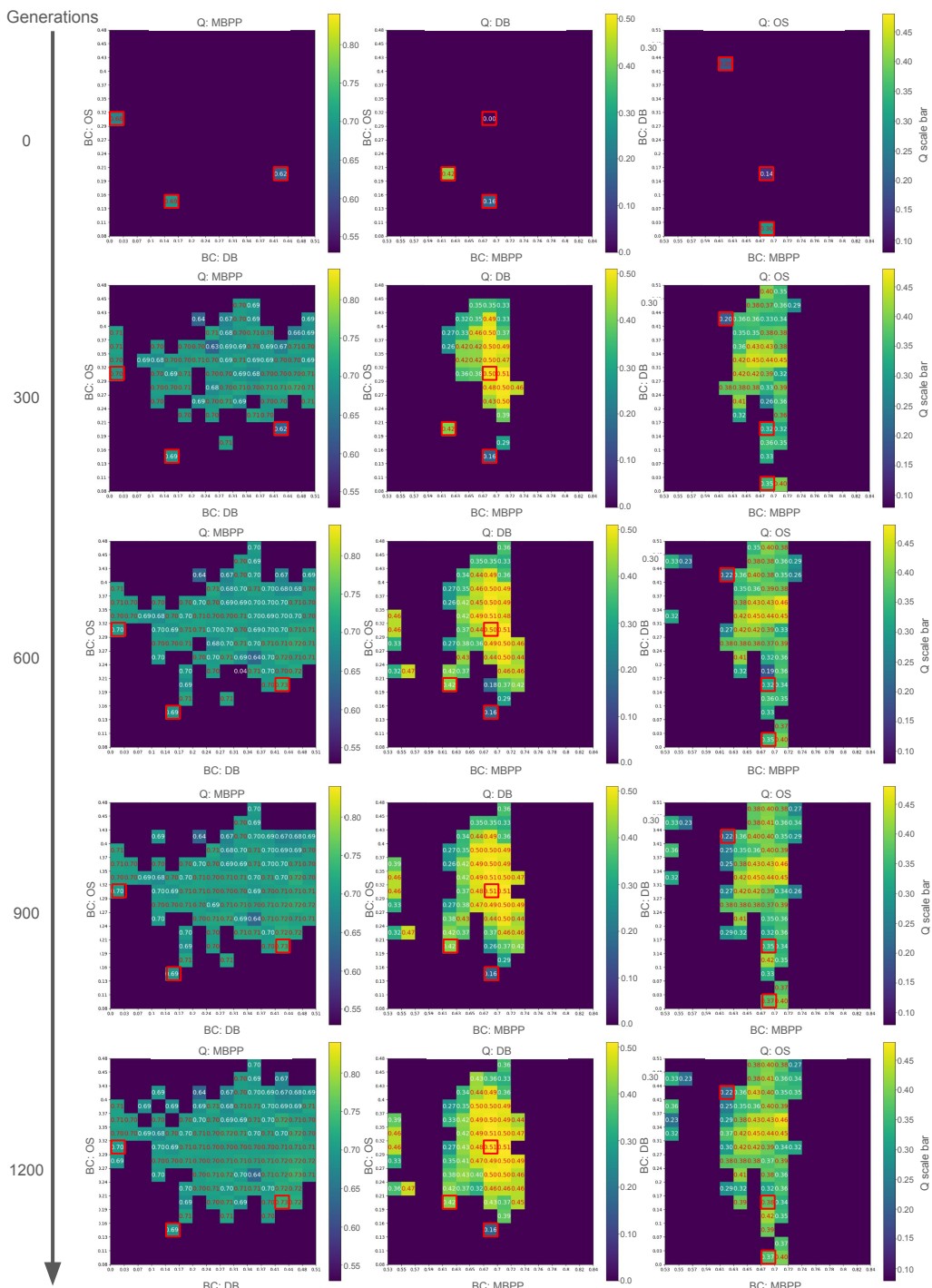

Figure 6: **CycleQD Development of Archives Across Generations.** In each archive, the two axes represent the BCs, and the color intensities in each grid indicate the quality of the LLM agent in that grid. The archives are shown in increments of 300 generations from top to bottom. The corresponding generation for each archive is displayed on the left side of the figure. The red bounding boxes indicate the grids where expert policies were present in each archive. The experiment shown in this figure is the same as in Figure 4, and the archive for 1200 generations is identical to that in Figure 4.

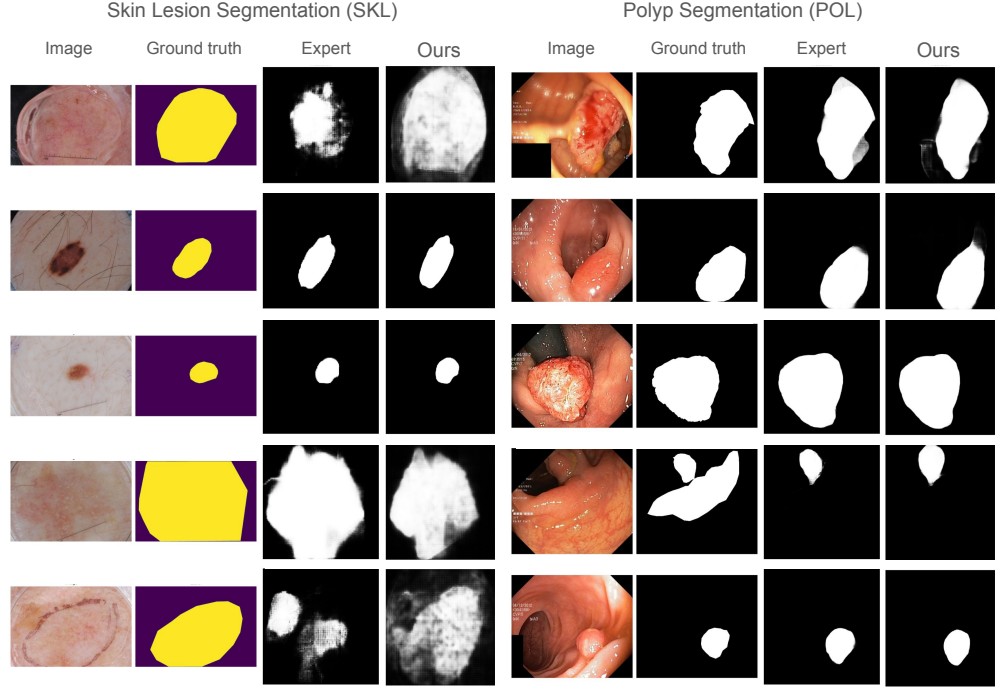

Figure 7: **CycleQD of SAM for SKL and POL.** Ours is a single merged model via CycleQD, and it is the same model as the one in Table 4 (# 3). The similarity between the two expert models is high (0.99), and the model maintains expert-level performance across both tasks.

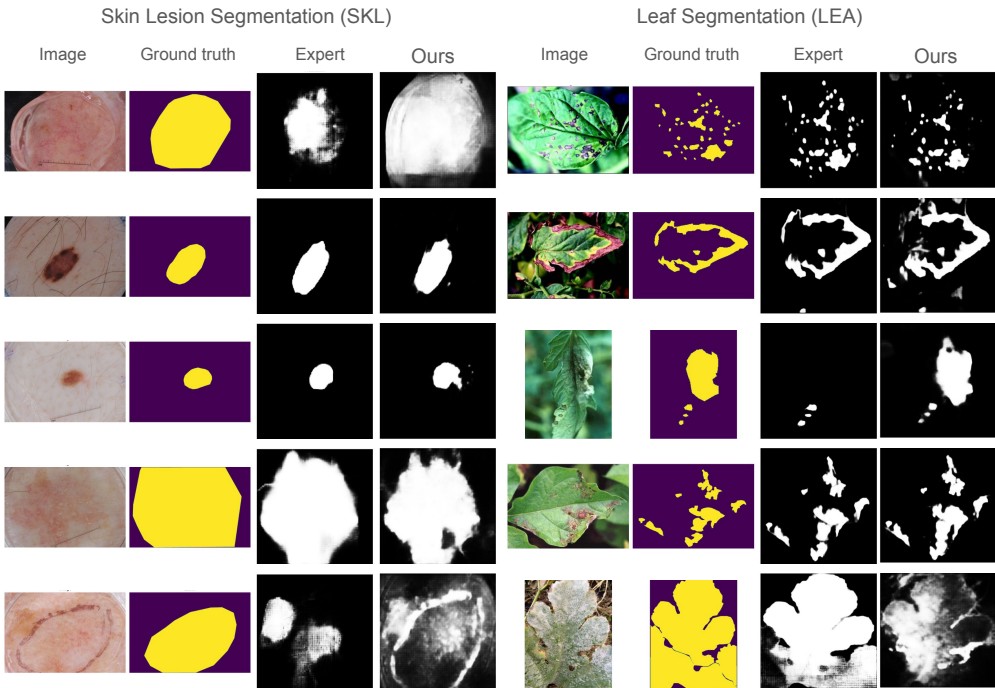

Figure 8: **CycleQD of SAM for SKL and LEA.** Ours is a single merged model via CycleQD, and it is the same model as the one in Table 4 (# 5). The similarity between the two expert models is 0.95, which is lower than #3 (0.99) from Table 4, resulting in a slight decrease in performance, as observed when comparing the bottom images in SKL.

