# OpenReview forum: "Agent Skill Acquisition for Large Language Models via CycleQD"
_ICLR.cc/2025/Conference — ICLR 2025 Poster_

### Official Review · Reviewer_kupr · 2024-10-28

**Soundness:** 3
**Presentation:** 3
**Contribution:** 3
**Rating:** 6
**Confidence:** 4

**Summary:**

This paper introduces CycleQD, a framework for enhancing large language models' skill acquisition through cyclical quality diversity optimization. CycleQD first trains expert models for individual skills, then combines them using model merging as crossover operations and SVD-based mutations, while cycling through different task metrics as quality measures. The method alternates between optimizing different skills by maintaining separate archives, each tracking models with diverse behavioral characteristics.

**Strengths:**

1. Demonstrates strong empirical results by achieving comparable performance to GPT-3.5-TURBO on coding, OS and DB tasks using only an 8B parameter model
2. Avoids the complex tuning of data ratios and objective functions by decomposing multi-skill training into single-skill training followed by model merging
3. Provides full ablation studies validating the necessity of key components including cyclic adaptation, SVD-based mutation, and elite sampling

**Weaknesses:**

1. Elite sampling strategy lacks theoretical foundation and uses arbitrary hyperparameters (0.5-0.8 range) without clear justification.
2. Computational costs and efficiency not analyzed or compared with baseline methods.
3. Missing comparison with established multi-objective optimization approaches like NSGA-II or Pareto-based methods.

**Questions:**

1. Why is cyclic optimization more effective than simultaneous optimization of multiple objectives?
2. What is the theoretical basis for using SVD decomposition in mutation, how does it help explore the parameter space more effectively?
3. Can this method scale to combining skills from vastly different domains (e.g., math, creative writing, and visual reasoning)? Would it still outperform traditional SFT?

---

> ### Author Response · Authors · 2024-11-23
> **Responses to Reviewer kupr (Part 1)**
>
> We thank the reviewer for taking the time to provide thoughtful comments and suggestions. We are happy to see that the reviewer agrees our work delivers strong empirical results and provides studies validating the design choices. The following are our responses to the comments and questions, please see the details in the revised PDF.
>
> **About the theoretical foundation of elite sampling and hyperparameter selection.**
>
> We have added text in Sec 3.1 to provide a foundation for elite sampling, and have clarified how the hyperparameters were selected in Sec A.1.3.
>
> In summary, previous research in QD explores sampling methods that expand the frontier not only in Q but also within BCs, allowing for more diverse behavior exploration and higher quality solutions. For example, the paper “Multi-objective Optimization-based Selection for Quality-Diversity by Non-dominated Sorting” evaluates whether each elite is on the Pareto frontier by checking if it is dominated within each behavior space and samples from elites on the frontier. Inspired by this work, we also incorporate behavioral space to sample elites, with two main differences from prior approaches. (1) Both Q and BCs are task performances and are treated equally in a cyclic process. (2) To achieve higher performance, pushing the frontier in a high-performance direction rather than a low-performance one is ideal. Therefore, we aggregate Q and BC values into a single metric to drive the frontier towards higher performance.
>
> Our hyperparameters are determined in an arbitrary manner: we manually picked a set of values, ran CycleQD for a small number of generations with each setting and picked the best. We believe more sophisticated methods such as grid search will generate better results.
>
> **About computation costs.**
>
> We have added a comparison between CycleQD and gradient-based fine-tuning in A.1.7. As a summary, the gradient-based fine-tuning took ~200 GPU hours, and CycleQD took ~410 GPU hours (excluding the experts training time). We wish to point out that (1) the extra time spent on CycleQD is mostly due to agentic task evaluations rather than the optimization, (2) due to GPU memory constraints, the merging process was carried out on CPU, (3) CycleQD produces an archive of models whereas FT generates only one. Furthermore, it is important to note that fine-tuning is a mature and well-established approach with software implementations that are optimized for efficiency. By contrast, the proposed method was developed primarily to assess its performance, and as such, there is ample room for further efficiency improvements.

---

> ### Author Response · Authors · 2024-11-23
> **Responses to Reviewer kupr (Part 2)**
>
> **About comparison with NSGA-II.**
>
> We have conducted a comparison with NSGA-II and have presented the setups and results in Sec 4.1. In summary, as is shown in the table below, while NSGA-II outperforms other model-merge-based baselines, it remains inferior to CycleQD. Additionally, it is worth noting that NSGA-II is known to have two limitations: (1) performance degradation as the number of objectives increases, and (2) reduced effectiveness of the crowding distance mechanism in high-dimensional spaces. CycleQD does not suffer from these limitations, further emphasizing the generality and robustness of our method.
>
> | #  | Methods                       | MBPP | DB   | OS   | Avg  |
> |----|-------------------------------|------|------|------|------|
> | 7  | Merging (w/o learning)        | 72.9 | 24.7 | **42.6** | 46.7 |
> | 8  | Merging (learning w/ GD)      | 69.3 | 41.2 | 29.6 | 46.7 |
> | 9  | Merging (learning w/ CMA-ES)  | 69.3 | 41.2 | 30.2 | 46.9 |
> | 10 | Merging (learning w/ NSGA-II) | 75.9 | **42.4** | 36.4 | 51.6 |
> | 11 | CycleQD (Ours)                | **76.4** | 38.2 | **42.6** | **52.4** |
>
>
> **About why cyclic optimization is more effective.**
>
> We share our insights as to why cyclic optimization is more effective in the following.
>
> For simultaneous optimization where aggregation of the objective functions are needed: as stated in the introduction, the tuning of task ratios in the objective function is often required, which can be challenging and highly problem-specific. Similar to coordinate descent, CycleQD effectively avoids this problem by using task metrics as both Q and BCs and cyclically optimize through each of them, ensuring balanced improvements without the need for explicitly weighted aggregation.
>
> For simultaneous optimization where objective aggregation is not necessary (e.g., NSGA-II): CycleQD’s cyclic optimization mechanism incorporates elements resembling non-dominated sorting. However, unlike NSGA-II, which handles all tasks simultaneously and relies on crowding distance for diversity, CycleQD alternates between tasks and naturally encourages diversity via the QD mechanism. These settings reduce interference between objectives and allows for deeper refinement of task performance.
>
>
> **About the theoretical basis for SVD mutation.**
>
> We have added texts at the end of Sec 3.3 to explain why SVD-based mutation.
>
> In summary, our SVD-based mutation provides two key advantages: (1) SVD focuses mutations on the fundamental building blocks of the task vector, which represent the essential components of agent skills. By constraining perturbations to these meaningful directions, we avoid the excessive exploration inherent in random perturbations, reducing the risk of overfitting. (2) The ability to manipulate individual components (i.e., $\omega$) offers finer control over $\theta_\texttt{child}$, enabling targeted modifications at a granular level.
>
>
> **About combining skills from vastly different domains and possibility to outperform SFT.**
>
> In addition to the three CS tasks, we have incorporated a VQA task into CycleQD in this revision (see A.1.5). As a vision-language modeling task, VQA differs significantly from computer science tasks, making this incorporation particularly challenging. During the CycleQD process, we extract the LLAMA3-8B-INSTRUCT component from LLAMA3-LLAVA-NEXT-8B and fuse it with other expert models.
>
> The results indicate that CycleQD achieves better VQA performance compared to its expert model, while maintaining higher performance on MBPP and OS than their respective experts. Although its DB performance is lower than the DB expert, CycleQD achieves the highest average performance overall.

---

> ### Author Response · Authors · 2024-12-02
> **Follow-up to Rebuttal**
>
> We would like to thank reviewer kupr once again for your time and efforts in reviewing our work.
>
> With our response and rebuttal revision, we made our best efforts to try and address the points raised in your constructive feedback. As the discussion period is coming to a close, we would really appreciate it if you could let us know if you found our response satisfactory and your thoughts on the revised manuscript.
>
> Please, do not hesitate to raise any further questions or concerns about our work.

---

### Official Review · Reviewer_8w66 · 2024-11-04

**Soundness:** 2
**Presentation:** 1
**Contribution:** 3
**Rating:** 6
**Confidence:** 3

**Summary:**

The paper proposed CycleQD to achieve superior performance on multiple agent skills while retaining robust language capabilities.

**Strengths:**

- The paper discussed a critical problem: achieving superior performance on multiple skills while retaining language capabilities.
- The proposed method is novel.

**Weaknesses:**

Overall, the major weakness of this paper is its presentation. For the audience not familiar with evolutionary computation, the paper is hard to follow.
1. [Presentation] Problems in understanding the proposed method.
	- Explain the evolution of figure 4 by referring to algorithm 1. The figures are meant to give direct intuitions about how the algorithm works, but the current delivery fails to achieve this purpose.
	- In the elite sampling (line 217), what does the formula intuitively means? Specifically, what is the meaning of $ \gamma$ .
	- In the crossover (Sec. 3.2), there are no explanations for why $w$ are sampled from gaussian? From my understanding, $w$ are used to determine component weights for combining task vectors. The component weights should be greater than $0$ . But Gaussian samples has infinite range.
	- In the mutation (Sec. 3.3), there are no explanations for why SVD can address the challenge of excessive exploration in parameter space or overfitting. In addition, clarify how the proposed SVD-based mutation allows finer control of $\theta_{child}$ (line 283).
	- In the model aggregation (Sec. 3.4), there are no explanations for why the proposed convex combination is adopted for obtain the final model. In particular, what is the connection between this design choice and the major goal of this paper, i.e., achieving superior performance on multiple tasks without degradation on other tasks.
2. [Contribution] CycleQD on SAM is irrelevant with the main topic of this paper, which focuses on language models.
3. [Soundness] The current experiments include only three tasks. Including more tasks to support the generality of CycleQD is better.

**Questions:**

1. Questions on presentation issues: see above.
2. Is it possible to combine finetuning and CycleQD, e.g., SFT after CycleQD? What would be the result.
3. Computation and time cost comparisons between CycleQD and gradient-based fine-tuning.
4. The number of generations is critical for selecting elites. How sensitive is CycleQD regarding this hyperparameter?

---

> ### Author Response · Authors · 2024-11-23
> **Responses to Reviewer 8w66 (Part 1)**
>
> We are grateful for the reviewers’ detailed feedback and suggestions. We are also delighted to see that the reviewer thinks our method to be a novel solution to a critical problem. The following are our responses to the comments and questions, please see the details in the revised PDF.
>
> **About the presentation of Algorithm 1.**
>
> We have added text to refer to Figure 4 (now Figure 6 after revision) while explaining Algorithm 1. In addition, we have also added a background section and Figure 2 to introduce evolutionary algorithms, specifically MAP-Elites, before describing our method to get the readers familiar with the operations mentioned in Algorithm 1. We hope this addresses the reviewer’s concern.
>
>
> **About the meaning of $\gamma$ and the intuition of elite sampling.**
>
> The meaning of $\gamma$ and its related components are detailed on lines 189 and 194, with the reasoning behind the design elaborated in the same paragraph. We have also added more text to provide a theoretical foundation for elite sampling. We hope this addresses the reviewer's concern.
>
>
> **About the sign of $\omega$ in the crossover operation.**
>
> The model mixing coefficients $\omega$ do not necessarily have to be positive numbers. Task vectors play an important role in model performance on certain tasks. By allowing negative component weights, the merged model has more freedom in optimizing its task vectors. And this optimization is conducted via evolutionary search. We have added this explanation in Sec 3.2 to clarify this.
>
>
> **About the explanation of SVD-based mutation and its finer control.**
>
> We have added texts at the end of Sec 3.3 to explain why SVD addresses this problem, and also clarify how SVD-based mutation allows finer control.
>
> In summary, our SVD-based mutation provides two key advantages: (1) SVD focuses mutations on the fundamental building blocks of the task vector, which represent the essential components of agent skills. By constraining perturbations to these meaningful directions, we avoid the excessive exploration inherent in random perturbations, reducing the risk of overfitting. (2) The ability to manipulate individual components (i.e., $\omega$) offers finer control over $\theta_\texttt{child}$, enabling targeted modifications at a granular level.
>
>
> **About model aggregation design.**
>
> As the reviewer pointed out, the purpose of this paper is to create a single model that shows consistently high performance across multiple tasks minimizing performance degradation between tasks. CycleQD generated numerous elite models specialized for each task. While these models have the potential to be applied to various use cases, we aggregated them to obtain a “generalist” model. To this end, a model aggregation method is needed.
>
> While there are numerous ways one can choose for model aggregation. We adopted a simple convex combination, which created the weights via softmax of the task performances. This is an arbitrary choice, and other methods (e.g., simple average of the performances) yielded an aggregated model of similar performance.

---

> ### Author Response · Authors · 2024-11-23
> **Responses to Reviewer 8w66 (Part 2)**
>
> **About the value of experiments on SAM.**
>
> The purpose of this experiment is to demonstrate the generality of our method across different domains, which we believe is an important aspect of the paper’s contribution.
>
> While the primary focus of our work is on merging LLMs to acquire agentic skills, one of the key strengths of CycleQD lies in its domain-agnostic design (i.e., task metric is the only thing that is required and it serves as both Q and BC). By applying CycleQD to SAM, we aim to prove this point and highlight that the proposed method is not limited to language models but can be generalized to other foundation models in the future for more diverse agentic skills.
>
>
> **About adding more tasks to support the generality of CycleQD.**
>
> In addition to the three CS tasks, we have incorporated a VQA task into CycleQD in this revision (see A.1.5).
>
> As a vision-language modeling task, VQA differs significantly from computer science tasks, making this incorporation particularly challenging. During the CycleQD process, we extract the LLAMA3-8B-INSTRUCT component from LLAMA3-LLAVA-NEXT-8B and fuse it with other expert models.
>
> The results indicate that CycleQD achieves better VQA performance compared to its expert model, while maintaining higher performance on MBPP and OS than their respective experts. Although its DB performance is lower than the DB expert, CycleQD achieves the highest average performance overall.
>
>
> **About combining CycleQD with SFT.**
>
> We have further fine-tuned the model optimized by CycleQD with the entire datasets for Coding, DB, and OS (model 11 in the following table). However, the fine-tuned model exhibits lower performance compared to the original CycleQD model (model 10). This suggests that CycleQD enhances the model's capability to its limit. Additionally, the fine-tuned model's performance is lower than that of other expert models (models 3, 4, 5, 6), indicating that it may have fallen into overfitting.
>
> | Method                         | MBPP |  DB  |  OS  | Avg  |
> |--------------------------------|------|------|------|------|
> | 3. Fine-tuning (Coding expert) | 70.4 | 21.2 | 20.7 | 37.4 |
> | 4. Fine-tuning (DB expert)     | 65.8 | 42.4 | 28.5 | 45.6 |
> | 5. Fine-tuning (OS expert)     | 66.3 | 0    | 30.4 | 32.2 |
> | 6. Fine-tuning (All)           | 67.3 | 37.1 | 36.7 | 47.0 |
> | 10. CycleQD (Ours)             | 76.4 | 38.2 | 42.6 | 52.4 |
> | 11. CycleQD + ALL              | 68.8 | 30   | 26.3 | 41.7 |
>
> **About computation comparison between CycleQD and gradient-based fine-tuning.**
>
> We have added this comparison in A.1.7. As a summary, the gradient-based fine-tuning took ~200 GPU hours, and CycleQD took ~410 GPU hours (excluding the experts training time). We wish to point out that (1) the extra time spent on CycleQD is mostly due to agentic task evaluations rather than the optimization, (2) due to GPU memory constraints, the merging process was carried out on CPU, (3) CycleQD produces an archive of models whereas fine-tuning generates only one. Furthermore, it is important to note that fine-tuning is a mature and well-established approach with software implementations that are optimized for efficiency. By contrast, the proposed method was developed primarily to assess its performance, and as such, there is ample room for further efficiency improvements.
>
>
> **About sensitivity to the number of generations.**
>
> We have added a sensitivity study to reveal this. Specifically, we tested CycleQD performance every 100 generations and added the results in A.1.3. In Figure 5, the performance gradually increases over the generations. We don’t observe large oscillations in performance w.r.t the number of generations.

---

> > ### Comment · Reviewer_8w66 · 2024-11-26
> > **Reviewer Response**
> >
> > Thank you for the rebuttal. I have raised the score.

---

### Official Review · Reviewer_mjB3 · 2024-11-05

**Soundness:** 3
**Presentation:** 3
**Contribution:** 3
**Rating:** 6
**Confidence:** 2

**Summary:**

This paper introduces evolutionary approaches for merging large language models (trained on individual skills), as an alternative to fine-tuning (jointly on multiple skills). This is the first work to merge models using Quality Diversity (QD) algorithms. This work introduces a cyclic adaptation for QD: the optimization is conducted for one task at a time (in a cycle). Models are merged in the crossover operation by linearly combining task vectors. Finally, the mutation function modifies the "child" (from crossover) by sampling perturbations from the SVD components of its task vector.

This approach is tested using coding, OS, and DB skills. This evolutionary approach for model merging outperforms both fine-tuning based methods (by +5.4 pts) and other model merging methods (by +5.5 pts). The ablations show that each component of this method (cycle adaptation, SVD-based mutation, and elite sampling) lead to gains (on average). This method also generalizes to out-of-distribution coding tasks (more than fine-tuned, single-task experts) while also retaining performance on language skills. On the other hand, the merged model produced by this approach underperforms on individual tasks compared to fine-tuned experts for the domain of image segmentation.

**Strengths:**

This work appears to be very original: (1) it is different from conventional fine-tuning and even model merging methods, and (2) it alleviates certain design decisions, like data mixing ratios and different objectives, when fine-tuning on multiple tasks.

I understand that previous evolutionary approaches don't update the weights directly and this method proposes to do so (using the framing of model merging). I think this is a significant contribution. I think the specific adaptations they propose are also positive contributions.

From what I can see, this method establishes sufficient gains over the alternative methods in the computer science tasks. Since this paper introduces several components, I was looking for (and was satisfied to find) the ablations in Table 3.

**Weaknesses:**

I think the writing is mostly clear, but this paper assumes some extra background knowledge about evolutionary methods. I would really appreciate it if the authors (briefly) introduce such approaches and terminology from first principles, before the details of their augmentations.

It would also be great if the authors could again motivate the need for evolutionary approaches (rather than fine-tuning) in training or merging models. Is it just the avoidance of design decisions that I mentioned above? Are there more fundamental motivations (like the diversity of skills captured by this method, which was briefly mentioned)? Or is this direction of work primarily exploratory?

I am not already familiar with evolutionary approaches, so no significant methodological weaknesses are obvious to me. I will wait to see what other reviewers have to say, but generally hold a positive opinion for now.

**Questions:**

Formatting: I see that a couple half-column figures are used (concatenated horizontally with text). Please make sure that this abides by the formatting rules and correct if not.

Is it normal for model merging approaches to test using just 2-3 tasks?

---

> ### Author Response · Authors · 2024-11-23
> **Responses to Reviewer mjB3**
>
> We appreciate the reviewers’ thoughtful comments and valuable insights. And we are grateful that the reviewer thinks our work to be very original. The following are our responses to the comments and questions, please see the details in the revised PDF.
>
>
> **About introducing background knowledge.**
>
> We have created a “Preliminaries” section (see Sec 2) to introduce the concepts in evolutionary algorithms and have added a figure to illustrate the flow. We specifically introduced concepts like crossover and mutation in the context of MAP-Elites to help paint the picture before diving into the details in Sec 3.
>
> **About the motivation for evolutionary approaches.**
>
> Our motivation is (1) to reduce the intricate design decisions and hyperparameter tuning, and (2) as an exploratory work, to show that evolutionary algorithms can be incorporated into LLM fine-tuning pipeline, which is currently dominated by gradient-based methods, as a compelling modification. We have added text in the introduction to emphasize (1), and (2) is stated on lines 94-95. We hope the added text would better clarify our motivation.
>
>
> **About formatting.**
>
> We reviewed [the ICLR 2025 submission template](https://www.overleaf.com/latex/templates/template-for-iclr-2025-conference-submission/gqzkdyycxtvt) and found no restrictions on using half-column figures. In addition, some accepted papers of ICLR 2024 use a half-column figure format (papers [1](https://openreview.net/attachment?id=eUgS9Ig8JG&name=pdf), [2](https://openreview.net/attachment?id=kmn0BhQk7p&name=pdf), [3](https://openreview.net/attachment?id=csukJcpYDe&name=pdf)). Based on this, we conclude that half-column figures are permissible.
>
>
> **About the number of tasks.**
>
> To show the general applicability and versatility of CycleQD, we have added a VQA task in addition to the three CS tasks (see A.1.5). One of the motivations for this work is to show that evolutionary strategies can be an effective component in the LLM post-training pipeline, and we believe that CycleQD has laid a foundation for future exploration with more tasks.

---

> > ### Comment · Reviewer_mjB3 · 2024-11-23
> >
> > Thanks to the authors for their response and clarifications. I have read their updated revisions and the other reviews/responses. I am satisfied with the improvements and author responses. I think this paper should be accepted. My true score would be a "7".

---

### Official Review · Reviewer_tJqK · 2024-11-08

**Soundness:** 3
**Presentation:** 2
**Contribution:** 3
**Rating:** 6
**Confidence:** 1

**Summary:**

This paper presents CycleQD, a novel framework for making llm acquire specific skills through Quality Diversity (QD) optimization. CycleQD addresses challenges in skill training, such as data imbalance and inadequate objective functions, by cyclically alternating quality measures for different tasks.  CycleQD shows competitive results, achieving performance on par with GPT-3.5-TURBO.

**Strengths:**

The paper presents an innovative approach to LLM skill acquisition, uniquely applying the Quality Diversity paradigm to cycle through task-specific optimizations, which is quite novel.
The experimental results on the proposed CycleQD framework show a substantial performance gain, validating the model’s effectiveness and overall performance.
Overall, CycleQD introduces a scalable method to merge agent skills into LLMs, addressing critical challenges in agent-based LLM design.

**Weaknesses:**

Although the CycleQD  is designed for the agent skill acquirment of LLMs, the experiments predominantly focus on computer science tasks, such as OS and DB. Its applicability to other fields as agents will futher strength this paper.

 The methodology, though effective, involves a multi-step process (crossover, mutation, cyclic quality alternation) that may be complex. The author may need to provide a clearer illustration of the methodology, such as an overview figure of the method pipeline.

The ablation study does not provide substantial evidence of the proposed method's effectiveness, as the performance gains of each component are minimal.

**Questions:**

I am also curious about any real-world application tests using the proposed method across different scenarios, given that the scenarios in the experiments are quite similar.

---

> ### Author Response · Authors · 2024-11-23
> **Responses to Reviewer tJqK**
>
> We thank the reviewer for the constructive feed and thinking of CycleQD as an innovative and scalable solution to merge agent skills. The following are our responses to the comments and questions, please see the details in the revised PDF.
>
> **About CycleQD’s applicability to other fields**
>
> In addition to the three CS tasks, we have incorporated a VQA task into CycleQD (see A.1.5).
>
> As a vision-language modeling task, VQA differs significantly from computer science tasks, making this incorporation particularly challenging. During the CycleQD process, we extract the LLAMA3-8B-INSTRUCT component from LLAMA3-LLAVA-NEXT-8B and fuse it with other expert models.
>
> The results indicate that CycleQD achieves better VQA performance compared to its expert model, while maintaining higher performance on MBPP and OS than their respective experts. Although its DB performance is lower than the DB expert, CycleQD achieves the highest average performance overall.
>
>
> **About a clearer illustration of the methodology**
>
> We have created a “Preliminaries” section (see Sec 2) to introduce the concepts in evolutionary algorithms and have added a figure to illustrate the flow. We also refer to this figure when introducing our CycleQD. We hope this describes our method clearer.
>
>
> **About ablation studies.**
>
> We have revised Sec 4.1.5 to better articulate the significance of our ablation studies and to highlight the following aspects.
>
> The cumulative improvement from baseline QD to our final CycleQD is substantial at 4.8 percentage points (from 47.6% to 52.4%). This is a meaningful improvement in the context of LLMs performing complex agent tasks, particularly considering our model approaches GPT-3.5-TURBO's performance despite having significantly fewer parameters.
>
> Each component shows clear contributions through carefully controlled experiments:
> - Introduction of CycleQD: +1.8% improvement over standard QD
> - SVD-based mutation: +2.3% improvement over no mutation
> - Elite sampling: +0.7% improvement over random sampling
>
> Most notably, our ablation studies reveal important qualitative insights about design choices. For instance, the comparison between mutation strategies (trials 1-3 in Table 2) demonstrates that naïve Gaussian mutation actually hurts performance (48.5% vs 49.4% with no mutation), while our SVD-based mutation shows clear benefits (51.7%). This validates the importance of our design choices and shows that improvements aren't merely additive - poor design choices can harm performance while our well-designed components work synergistically.
>
>
> **About real-world applications across different scenarios.**
>
> We have added a VQA task to demonstrate the general applicability of our method beyond the original CS tasks, highlighting the versatility of our approach across different scenarios. At the same time, we acknowledge the challenges of deploying an 8B LLM in real-world scenarios. While smaller models are computationally more feasible for experimentation, their capabilities are often limited compared to larger LLMs. Nonetheless, our work demonstrates that evolutionary strategies can be an effective component in the LLM post-training pipeline, laying a foundation for future exploration with more capable models.

---

> > ### Comment · Reviewer_tJqK · 2024-11-26
> >
> > Thanks to the authors for their response and clarifications. I have reviewed their revisions and the other reviews.

---

### Author Response · Authors · 2024-11-23
**General Response to Reviewers and AC**

We sincerely thank all the reviewers for their time and effort in evaluating our paper, as well as for their constructive and insightful feedback. We have carefully considered the comments and questions provided, and have made revisions to address these concerns. Additionally, we have conducted new experiments and updated the manuscript to reflect these changes (All changes are highlighted in blue text in the revised PDF).

Below, we provide a summary of the modifications made:
**Writing**
- Added a “Preliminaries” section before the “Methods” section (see Sec 2), repositioned the “Related Works” section to a later part of the paper (see Sec 5), and relocated most of its content to the appendix due to space limitations (see Sec B).
- Added an illustration figure in the “Preliminaries” section. (see Sec 2)
- Added text to highlight the significance of our ablation studies (see Sec 4.1.5)
- Added text to highlight our motivation (see Sec 1)
- Expanded writing in the “Methods” section to refer to figure 4 (now figure 6 in the revised PDF) while introducing Algorithm 1 (see Sec 3)
- Added text to explain why $\omega$ do not have to be positive (see Sec 3.2)
- Added text to clarify how SVD-based mutation addresses excessive exploration and how it allows finer control (see Sec 3.3)
- Rephrased text to emphasize the necessity of model aggregation (see Sec 3.4)
- Added computation time comparison (see A.1.7)
- Added text to explain the design decision behind elite sampling (see Sec 3.1) and clarified our hyperparameters choice (see A.1.3)

**Experiments**
- Added a VQA task in addition to the three CS tasks in CycleQD (see A.1.5)
- Added a CycleQD+SFT experiment (see response to reviewer 8w66)
- Added a sensitivity study (see A.1.3)
- Added a comparison with NSGA-II (see Sec 4.1.4)

We hope that our responses and revisions satisfactorily address the reviewers’ concerns. If so, we kindly ask the reviewers to consider re-evaluating their ratings in light of these improvements.

---

### Meta-Review · Area_Chair_k2ww · 2024-12-20

**Metareview:**

This paper introduces CycleQD, a framework for skill acquisition in large language models (LLMs) using Quality Diversity (QD) optimization. By cyclically alternating quality measures for different tasks, CycleQD offers a scalable alternative to fine-tuning and traditional model merging methods, addressing challenges such as data imbalance and inadequate objective functions. The proposed method demonstrates strong performance, achieving gains over fine-tuning and other model merging techniques, particularly on coding, OS, and database tasks, while also generalizing to out-of-distribution tasks. Reviewers commend the originality of applying QD algorithms to LLMs and the clear performance improvement. However, the paper primarily focuses on computer science tasks, limiting its demonstrated applicability to other domains. Additionally, the methodology could benefit from more precise illustrations and more substantial ablation evidence. While the paper is well-received for its novelty and contributions, reviewers suggest further exploration of real-world applications, broader task coverage, and more accessible explanations of evolutionary methods.

**Additional Comments On Reviewer Discussion:**

The authors' rebuttal convinced the reviewers, and some of them raised a positive score.

---

### Decision · Program_Chairs · 2025-01-22

Accept (Poster)